# Characteristics of the Perianthic Endophytic Fungal Communities of the Rare Horticultural Plant *Lirianthe delavayi* and Their Changes under Artificial Cultivation

**DOI:** 10.3390/microorganisms12071491

**Published:** 2024-07-21

**Authors:** Lang Yuan, Tongxing Zhao, Jing Yang, Nannan Wu, Pinzheng Zhang, Hanbo Zhang, Tao Xu

**Affiliations:** 1School of Ecology and Environmental Science, Yunnan University, Kunming 650106, Chinayangjing202306@126.com (J.Y.);; 2State Key Laboratory for Conservation and Utilization of Bio-Resources in Yunnan, Yunnan University, Kunming 650106, China

**Keywords:** perianthic endophytic fungal diversity, artificial planting, wild, ornamental plant, *Lirianthe delavayi*

## Abstract

Flower endophytic fungi play a major role in plant reproduction, stress resistance, and growth and development. However, little is known about how artificial cultivation affects the endophytic fungal community found in the tepals of rare horticultural plants. In this research, we used high-throughput sequencing technology combined with bioinformatics analysis to reveal the endophytic fungal community of tepals in *Lirianthe delavayi* and the effects of artificial cultivation on the community composition and function of these plants, using tepals of *L. delavayi* from wild habitat, cultivated campus habitat, and cultivated field habitat as research objects. The results showed that the variety of endophytic fungi in the tepals of *L. delavayi* was abundant, with a total of 907 Amplicon sequencing variants (ASVs) obtained from all the samples, which were further classified into 4 phyla, 23 classes, 51 orders, 97 families, 156 genera, and 214 species. We also found that artificial cultivation had a significant impact on the community composition of endophytic fungi. Although there was no significant difference at the phylum level, with *Ascomycota* and *Basidiomycota* being the main phyla, there were significant differences in dominant and unique genera. Artificial cultivation has led to the addition of new pathogenic fungal genera, such as *Phaeosphaeria*, *Botryosphaeria*, and *Paraconiothyrium*, increasing the risk of disease in *L. delavayi*. In addition, the abundance of the endophytic fungus *Rhodotorula*, which is typical in plant reproductive organs, decreased. Artificial cultivation also altered the metabolic pathways of endophytic fungi, decreasing their ability to resist pests and diseases and reducing their ability to reproduce. A comparison of endophytic fungi in tepals and leaves revealed significant differences in community composition and changes in the endophytic diversity caused by artificial cultivation. To summarize, our results indicate that endophytic fungi in the tepals of *L. delavayi* mainly consist of pathogenic and saprophytic fungi. Simultaneously, artificial cultivation introduces a great number of pathogenic fungi that alter the metabolic pathways associated with plant resistance to disease and pests, as well as reproduction, which can increase the risk of plant disease and reduce plant reproductive capacity. Our study provides an important reference for the conservation and breeding of rare horticultural plants.

## 1. Introduction

Plants often harbor endophytic fungi in their tissues, which have complex relationships with plant hosts as important ecological niches for many microorganisms. Endophytic fungi are present in all plants that have been studied to date [1] and are distributed in the roots, stems, leaves, flowers, fruits, and seeds of plants [2]. Endophytic fungal diversity varies widely with environmental conditions [3], and even within the same plant species, the composition of endophytic fungal communities within tissues can change due to differences in the physiological states of individual plants [4]. Additionally, factors such as plant secretions [5], age, climate [6], nutritional balance, geographic location [7], and season [8] also influence the distribution of endophytic fungal communities. Other factors influencing the endophytic fungal community in flowers include flowering time [9], nectar [10], and pollinator vectors [11]. It is evident that many factors influence the variety of endophytic fungi in plants, and the factors affecting endophytic fungi in flowers are particularly complex and diverse.

As an important reproductive organ of plants, flowers harbor a rich, diverse, and dynamic microbial community [12]. Flowers provide growth and protection sites for microorganisms [13,14], and microorganisms play a crucial role in plant reproduction. Research on the role of microorganisms in plant reproduction has focused mainly on the study of microorganism communities in floral nectar. Floral nectar, as an important food source for many animal pollinators, attracts these pollinators and facilitates plant reproduction through pollination by animals. To the best of our knowledge, we have found that some yeasts play an integral role in the plant reproduction process. For example, Metschnikowia reukaufii (Ascomycota) can increase plant attraction to pollinators such as bees [15]. Some yeasts [16] can inhibit pollen germination by reducing pollen viability and causing pollen tube rupture. In addition, many studies [17,18] have shown that yeasts have a direct or indirect impact on plant reproduction. Therefore, it remains to be explored whether the flowers of plants harbor yeasts that are related to reproduction.

The plants in the *Magnoliaceae* family are dioecious and have different flowering periods, leading to a weak reproductive capability in natural environments [19], resulting in the endangered status of many species in the *Magnoliaceae* family. To address this situation, studying the floral microbiome of *Magnoliaceae* plants may be a viable option. It is generally believed that outer floral organs, such as petals and sepals, primarily attract pollinators [20], so the impact of endophytic fungi in petals on plant reproduction is often overlooked. Based on previous knowledge of endophytic fungi in the *Magnoliaceae* family [21,22,23], we have found that there is limited detailed research on endophytic fungi in the tepals of *Magnoliaceae* plants. *Lirianthe delavayi* is a member of the genus Lirianthe of the family *Magnoliaceae* [24]. It is highly valued for its medicinal and horticultural value [25] and has a long history of artificial cultivation.

During artificial cultivation, fertilizers, pesticides, and herbicides [26,27] can affect the endophytic fungi of plants, and the microbial community plays a profound role in helping plants resist pathogens and environmental stress. Hassani [28] reported no significant difference in the leaf fungal community between wild and cultivated wheat, whereas Abdullaeva [29] reported that root fungi of wild and cultivated wheat were less resistant to other pathogens. Derr [30] reported that endophytes in wild and cultivated *Agave* leaves were more influenced by geography and that artificial cultivation had no significant effect on leaf endophytes. However, in our previous investigation of endophytes in the leaves of *L. delavayi* [31], we found that artificial cultivation resulted in a significant reduction in endophyte diversity and an increase in the presence of pathogens such as *Alternaria*, which may increase the risk of plant diseases and pests. This finding is consistent with previous research [32]. There are also relevant studies on other ecological niches of plants in addition to the effects of cultivation on leaf endophytic fungi. For example, an investigation into root fungal communities on cultivated and wild rice [33] revealed that cultivation increased rice fungal diversity and that wild rice exhibited greater resilience and resistance to pathogen attack. However, it remains unknown whether artificial cultivation has an impact on endophytic fungal diversity in the tepals of *L. delavayi* and what impact it may have on the functional aspects of the endophytic fungal community. Moreover, leaves and flowers occupy similar ecological niches, but due to the different functions of plant reproductive and nutritional organs, comparing the endophytic fungal communities in leaves and petals will provide more persuasive evidence for exploring the effects of artificial cultivation on the diversity, function, and composition of endophytic fungi.

In summary, this report compares and analyzes the endophytic fungi within tepals of wild *L. delavayi* (W), horticulturally cultivated *L. delavayi* (CX), and naturalized cultivated *L. delavayi* (CY) using high-throughput sequencing technology. Based on our previous research on endophytic fungi within *L. delavayi* leaves, we hypothesize the following regarding the endophytic fungi within *L. delavayi* floral tepals: (i) artificial cultivation reduces the diversity and community composition of endophytic fungi within *L. delavayi* floral tepals; (ii) artificial cultivation affects the function of the endophytic fungi within the tepals.

## 2. Experimental Methods and Materials

### 2.1. Sample Profile

An investigation revealed that wild *L. delavayi* in Yunnan Province was mainly distributed in the central region, and Kunming and its surrounding areas belonged to the central part of Yunnan Province, where both wild and cultivated varieties of *L. delavayi* were found. To exclude the influence of weather conditions on the endophytic fungi of *L. delavayi*, we selected nearby areas for sampling; at the same time, to ensure data accuracy, each group of samples was collected with three or more biological replicates. In May 2023, we collected petal samples from *L. delavayi* in Yiliang, Kunming (103°11′50″ E, 25°14’12″ N), where we collected three groups of wild *L. delavayi* tepal samples (designated “W” hereafter). The surrounding habitat of *L. delavayi* in this habitat is rich in various other wild plants, grows vigorously in an undisturbed environment, and forms abundant vegetation (Appendix A). Moreover, in another area of Yiliang, Kunming (103°19′48″ E, 25°20’97″ N), we collected 12 groups of cultivated *L. delavayi* plants grown in a wild environment (designated “CY” hereafter). The main characteristic of this habitat was orderly plant cultivation, with significant human activity influencing the growth environment. Finally, we collected 16 groups of horticulturally cultivated *L. delavayi* tepal samples on the Yunnan University campus (102°84′92″ E, 24°82′75″ N) (designated “CX” hereafter). In this habitat, various flowers, grasses, and trees were clearly visible on the campus, and the growth environment was clean and tidy. Most importantly, *L. delavayi* plants grown in this environment are often subjected to artificial management, such as fertilization and watering (Appendix A). This habitat still exhibited unique ecological features (Appendix A). Each group was sterile, and bagged and stored at 4 °C. The samples were processed within 24 h after collection. A total of 31 groups of samples were collected (Appendix A).

For pretreatment of the collected samples, healthy tepals were selected, rinsed with tap water, and dried. The tepal samples were then subjected to the following procedures: 2% sodium hypochlorite for 2 min, 75% ethyl alcohol for 2 min, three sterile water rinses, and sterile filter paper to dry the surface. Sterile scissors were used to cut the samples into small 0.1 × 0.1 cm pieces, which were subsequently placed in sterile plastic tubes and stored at −80 °C until they were used for microbial analysis.

### 2.2. DNA Extraction, PCR Amplification, and High-Throughput Sequencing

Initially, the CTAB method was used to extract total DNA from the microbial community of the tepal samples. Agarose gel electrophoresis followed by quantification using a UV spectrophotometer was used to assess the quality of the extracted DNA. Subsequently, the ITS2 (internal transcribed spacer 2) region of fungal DNA was amplified using the primers ITS1 and ITS4 [34]. The PCR amplification protocol was as follows: predenaturation at 94 °C for 1 min, 30 cycles of denaturation at 94 °C for 30 s, and baking at 50 °C for 45 s, with a final extension of 10 min at 72 °C. An AMPure XT kit (Beckman Coulter Genomics, Danvers, MA, USA) was used to detect the purified PCR products by 2% agarose gel electrophoresis. The amplification products were quantified and pooled for sequencing using an Illumina library quantification kit (Kapa Biosciences, Woburn, MA, USA). An Agilent 2100 Bioanalyzer (Agilent, Santa Clara, CA, USA) was used. The concentration of the qualified library should be greater than or equal to 2 nM. These libraries were then sequenced on a NovaSeq 6000 sequencer using 2 × 250 bp paired-end sequencing according to the NovaSeq 6000 SP reagent kit (500 cycles) after gradient dilution and NaOH denaturation to obtain single strands. After sequencing, raw offline data were obtained and then overlapped, quality controlled, and chimerism removed for clean data. To obtain representative sequences with single-base accuracy and to generate the final Amplicon sequencing variant (ASV) table and sequence numbers, the DADA2 (Amplicon Denoising Algorithm) [35] was used.

Lianchuan Biotechnology was responsible for the purity and integrity of the PCR products, PCR amplification, and Illumina MiSeq sequencing. The sequencing data were deposited in the NCBI Sequence Read Archive as PRJNA1109368.

### 2.3. Data Analysis

We concentrated on exploring fungal community differences between wild, horticulturally cultivated, and naturalized cultivated *L. delavayi*, as well as the impact of cultivation on endophytic fungi in tepals. For this reason, the samples from each tree were treated as replicates in the statistical analysis. The representative sequences of each ASV were classified and identified using BLAST with a confidence level of 0.8, identification threshold of ≥90%, query coverage of ≥80%, and e-value of ≤10^−5^ [36]. A summary of 907 ASVs was obtained, providing classification information at multiple taxonomic levels for all the samples. To investigate how artificial cultivation affects fungal community richness and evenness in *L. delavayi* tepals, a Venn [37] diagram was constructed using the Lingbo Microclass Cloud Platform (http://www.cloud.biomicroclass.com) (accessed on 3 June 2024) to display the fungal communities in the three groups of samples. Furthermore, alpha diversity indices were calculated using Mothur v.1.13.0 [38] and visualized using Origin software v.10.15 to demonstrate the similarity of endophytic fungi among the three sample groups. Additionally, beta diversity analysis was conducted using the principal coordinate analysis (PCoA) [39] method on the Lianchuan Biotechnology Cloud Platform (https://www.omicstudio.cn/tool, accessed on 3 June 2024) to explore the differences in endophytic fungal composition among the three groups, and a heatmap was constructed to illustrate the differences in fungal composition at the phylum and genus levels and to investigate the effect of horticultural cultivation on the endophytic fungal community. A discriminant analysis effect size (LEfSe) algorithm was used to detect taxa with distinctly variable relative abundances among the three groups, where taxa identified with LDA values > 3.5 were considered LEfSe biomarkers [40]. The FUNGuild database [41] was used for the prediction of the ecological functional guilds of fungi, and PICRUSt2 software (version picrust2.2.0b) was used for the prediction of fungal functions to study the effect of cultivation on the functional roles of endophytic fungi in plants. Subsequently, the functional annotation results obtained from the KEGG database were analyzed using the STAMP for variance analysis [42]. Specifically, we focused on annotating the functional pathways [43,44] at three levels to facilitate analysis and results presentation.

## 3. Results

### 3.1. Endophytic Fungi Diversity in Tepals of L. delavayi

From all the samples, a total of 907 ASVs were identified, which were further classified into 4 phyla, 23 classes, 51 orders, 97 families, 156 genera, and 214 species. Among them, the number of ASVs obtained from CY was the highest at 423, with 255 unique ASVs in this group. Next, for CX, 356 ASVs were obtained, including 186 unique ASVs. We had the fewest number of ASVs at 128, with 42 unique ASVs. There were 64 ASVs shared by the three groups (Figure 1a). Differential analysis based on the Shannon index revealed that endophytic fungal diversity in *L. delavayi* tepals was similar among the different samples (Figure 1b). However, the PCoA results indicated that significant differences were observed in the endophytic fungi sampled from different locations (Figure 1c), although there were some similarities in the endophytic fungal community and structure between the wild and cultivated samples.

### 3.2. Fungi Species Composition

At the phylum level, endophytic fungi in *L. delavayi* tepals from different regions were predominantly *Ascomycota*, followed by *Basidiomycota*, while *Zygomycota* and the unclassified phylum of endophytic fungi were the least abundant. Interestingly, members of *Zygomycota* were found in only a single sample from the naturalized cultivation, while the other samples did not show the presence of this endophytic fungus (Figure 2a).

At the genus level, apart from undefined *Ascomycota* groups and some other unknown genera that were common to all samples, there were distinct dominant and unique genera in wild and cultivated samples from different locations. In the CX, the dominant genera were *Asterotremella*, *Davidiella*, *Candida*, and *Kabatiella*, while the unique genera included *Ganoderma*, *Hannaella*, *Phaeosphaeria*, unclassified *Pleosporaceae*, *Botryosphaeria*, and *Paraconiothyrium*. In CY, the dominant genera were unclassified *Pleosporales*, *Malassezia*, and *Alternaria*, while the unique genera included unclassified *Lecanorales* and *Devriesia*. The dominant genera in W were *Dematiaceae*, *Asterotremella*, *Rhodotorula*, and *Alternaria*, with no unique genera identified. However, the abundances of *Dematiaceae* and *Rhodotorula* in the cultivated samples were extremely low. These findings suggest that the cultivated campus samples had a greater number of dominant and unique genera, followed by the cultivated wild samples, while the cultivated wild samples had a greater number of dominant genera but lacked unique genera (Figure 2b).

### 3.3. LEfSe Analysis

To further determine significant differences in endophyte abundance in wild and cultivated *L. delavayi* tepals in different environments and to explore community composition from a different perspective, linear discriminant analysis effect size (LEfSe) analysis was used to detect biomarkers at the phylum to genus level. Through LEfSe analysis, specific biomarkers of endophytic fungi in different environments were identified. At the genus level, endophytic fungi belonging to the genera *Asterotremella*, *Candida*, and *Phaeosphaeria* were found to be enriched in CX. In contrast, the genera *Torula*, *Alternaria*, *Phaeosphaeriopsis*, *Biatriospora*, *Peltaster*, and *Devriesia* were enriched in CY. Additionally, the genera *Coprinopsis*, unclassified *Mycosphaerellaceae*, *Cladosporium*, and *Periconia* were enriched in the wild samples. Linear discriminant analysis (LDA) (LDA > 3.5) revealed that *Asterotremella*, *Alternaria*, and *Cladosporium* had the most significant impacts on the cultivated, cultivated, cultivated, and cultivated campus samples, respectively (Figure 3).

### 3.4. Functional Prediction

#### 3.4.1. Functional Annotation of FunGuild

When annotating the functional profiles of endophytic fungi in the tepals of *L. delavayi*, these fungi were classified into three categories based on their nutritional modes: pathotrophs, symbiotrophs, and saprotrophs. After annotating endophytic fungi in the tepals of wild and cultivated specimens from two different regions, it was found that the main nutritional types were the pathotrophic–saprotrophic mixed type, pathotrophic–saprotrophic–symbiotic mixed type, saprotrophic nutritional type, and saprotrophic–pathotrophic mixed type. Among the cultivated specimens, in addition to the pathotrophic nutritional type showing a significant growth trend, the pathotrophic–saprotrophic mixed type, pathotrophic–saprotrophic–symbiotic mixed type, symbiotic nutritional type, and pathotrophic-symbiotic mixed type all showed growth trends. Compared with those in W, the abundance of the saprotrophic nutrient type in CX significantly increased, while that in CY did not increase (Figure 4).

Further annotation of functional groups revealed significant growth trends in cultivated specimens for animal pathogen–plant pathogen-unclassified saprotrophs, undefined saprotrophs, plant pathogens, endophytic fungus–plant pathogen-undefined saprotrophs, animal pathogens, parasitic fungus-undefined saprotrophs, endophytic fungus-lichen parasitic fungus–plant pathogen-undefined saprotrophs, and animal pathogen-undefined saprotrophs. A decreasing trend was observed for animal endophytic fungi, animal pathogens, symbiotic fungi, plant pathogens, defined saprotrophs, and coprophilous fungi, plant saprotrophic fungi, and wood saprotrophic fungi (Figure 5). Most types related to pathogens and saprotrophs showed an increasing trend. Animal pathogens were not present in the wild samples but exhibited high abundance in the cultivated samples, while plant pathogens had low abundance in the wild samples but showed a significant increase in the cultivated specimens.

#### 3.4.2. STAMP Differential Analysis

Based on the high-throughput sequencing results, we first predicted the metabolic pathways of endophytic fungi in the tepals using PICRUSt2 and then conducted a statistical analysis of the metabolic pathways using STAMP. According to the KEGG database, a total of 86 metabolic pathways (level 3) were predicted for the endophytic fungi present in the tepals. By comparing the metabolic pathways of different groups of fungi, we identified pathways with significant intergroup differences. To determine the exact impact of artificial cultivation on plant functionality, we compared all cultivated samples with wild samples. Additionally, to further ascertain the specific effects of human intervention on a cultivated species, we compared horticultural cultivars with wild-acclimated cultivars.

Comparison of the metabolic pathways between wild and cultivated species revealed a significant impact of artificial cultivation on *L. delavayi* metabolic pathways, with a total of 19 pathways showing statistical significance. These pathways include the de novo biosynthesis of adenosine nucleotides, the superpathway of de novo biosynthesis of adenosine nucleotide I, the biosynthesis of phosphatidylserine I, the superpathway of L-serine biosynthesis, the urea cycle, the biosynthesis of coenzyme A, the superpathway of pyrimidine ribonucleoside degradation, and starch degradation. In addition to the pyrimidine ribonucleoside degradation pathway, which was considerably greater in the wild species than in the cultivated species, all the other pathways were considerably greater in the cultivated population than in the wild population (Figure 6).

In the comparison between CX and CY, a total of nine metabolic pathways showed statistically significant changes. Aerobic respiration I (cytochrome c), mitochondrial fatty acid beta-oxidation, methyl ketone biosynthesis, and the Calvin cycle were significantly greater in wild-acclimated cultivars than in horticultural cultivars. Conversely, the remaining pathways—L-leucine degradation I, the superpathway of GDP-mannose-derived O-antigen building block biosynthesis, pyrimidine deoxyribonucleoside phosphorylation, de novo biosynthesis of pyrimidine deoxyribonucleoside I, and the superpathway of pyrimidine ribonucleoside degradation—showed the opposite trend (Figure 7).

## 4. Discussion

### 4.1. Effects of Artificial Cultivation on Endophytic Fungal Diversity in Tepals of L. delavayi

This study analyzed for the first time the impact of artificial cultivation on endophytic fungi in *L. delavayi* tepals. Our analysis results differed from our hypothesis, as the alpha diversity of endophytic fungi in W, CX, and CY was similar. Interestingly, this trend was consistent with previous laboratory findings on endophytic fungi in the leaves of *L. delavayi*. Notably, previous studies on leaf endophytic fungi indicated a decrease in plant endophytic fungal diversity due to cultivation, whereas the opposite trend was observed for tepal endophytic fungi. This discrepancy aligns with the findings of Coleman-Derr [30], suggesting significant differences in microbial diversity among different ecological niches. Cultivation can both decrease and increase microbial diversity in plants. Coincidentally, previous studies in the laboratory on endophytic fungi in leaves [31] have shown that artificial cultivation can lead to reduced diversity of endophytic fungi in plant leaves. However, our results show the opposite situation, suggesting that this may be due to antagonistic substances, such as floral scents [45] or floral secretions [46,47], which reduce the abundance of endophytic fungi in the petals compared to that in the leaves. Changes in endophytic fungal communities may be influenced by various environmental and host-related factors. Since microbial community assembly is based on ecological processes resulting from plant selection effects and environmental factors [48], we believe that the differential impact of cultivation on endophytic fungal diversity in different plant organs may be due to plant selection or environmental factors.

Considering the significant differences between artificial cultivation environments and natural environments for wild growth, we believe that artificial cultivation has a significant impact on the diversity of endophytic fungi in tepals. Therefore, the PCoA results indicate significant differences in species composition among the endophytic fungal communities of W, CX, and CY. Moreover, W and CY exhibited broader ASV variations than did CX, reflecting the significant environmental heterogeneity of W and CY.

### 4.2. Artificial Cultivation Leads to an Increase in Pathogenic Fungi and a Decrease in Beneficial Fungi within Tepals

All the samples showed a dominant phylum belonging to *Ascomycota* and *Basidiomycota*, with a relatively high proportion of *Ascomycota* (60–70%) and a relatively low proportion of *Basidiomycota* (20–30%). This finding is similar to those of other studies on flower microbiomes [49]. At the genus level, we found significant variations in dominant and unique genera among W, CX, and CY, with cultivation leading to a more diverse composition of endophytic fungi in tepals. We found that most of these dominant and unique genera are pathogenic fungi. For example, in the CX, genera such as *Davidiella* [50], *Candida* [51], *Kabatiella* [52], *Phaeosphaeria* [53], *Botryosphaeria* [54], and *Paraconiothyrium* [55] were predominant. In CY, genera such as *Malassezia* [56], *Alternaria* [50,57], and *Devriesia* [58,59] were predominant. In W, genera such as *Cladosporium* [60] and *Alternaria* were predominant. Most of these genera contain pathogenic strains that can cause various diseases in plants or animals. Moreover, LEFSe analysis revealed that enriched genera were mostly present in dominant and unique genera, such as *Candida*, *Torula*, *Alternaria*, and *Cladosporium*. Our results indicate that artificial cultivation obviously alters the community composition of plant endophytic fungi, particularly by adding new pathogenic genera, which may increase the risk of plant diseases.

Interestingly, the genus *Alternaria* is a typical endophytic pathogenic fungus that colonizes many plants [29,30]. However, our results show that *Alternaria* is not a dominant genus in CX but is dominant in W and CY. We found that plant pathogenic fungi have antagonistic effects on each other. In CX, the abundance of new pathogenic genera colonizing *L. delavayi* tepals may strongly inhibit *Alternaria*. However, the results exhibited by the *Alternaria* differ significantly from previous laboratory studies on endophytic fungi in leaves. Through our understanding, numerous contradictory pieces of evidence regarding the impact of artificial plant cultivation on microbial diversity suggest that potential artificial cultivation syndrome mitigation depends on plant species [61,62]. Moreover, studies have shown that plants likely control the assembly of fungal microbial communities as a protective mechanism against fungal pathogens [63,64]. Based on these findings, we believe that the significant difference in endophytic fungi between the nutrient organs and reproductive organs of *L. delavayis* may be the result of plant selection effects and limited fungal dispersal [28]. Additionally, the genus *Rhodotorula* is worth noting. Our analysis revealed a significant decrease in *Rhodotorula* during artificial cultivation, although it is a common floral yeast [65] that is crucial for attracting insects for pollination. In addition, *Rhodotorula* exhibits strong stress resistance [66], which suggests that cultivation may reduce the fertility and stress resistance of *L. delavayi*.

With regard to the functional annotation of the endophytic fungi in the tepals of *L. delavayi* on the basis of their nutritional type, we found that saprotrophic nutrition, pathotrophic–saprotrophic mixed nutrition, pathotrophic–saprotrophic–symbiotic mixed nutrition, and pathotrophic–saprotrophic mixed nutrition were the main types, followed by other mixed types, and symbiotic nutrition was the least common. This finding is consistent with studies by Zhou [67,68] and Wang [69] on the ecological functions of endophytic fungi in plant roots and flowers. The predominance of saprotrophic nutrients and their variants in the tepals of *L. delavayi* may be due to the short life cycle of the flowers and the need to promote the senescence of the tepals. A significant increase in pathotrophic nutrition, especially for plant and animal pathogenic fungi and their variants, was observed in the cultivated population. We also observed significant changes in the abundances of undefined saprotrophs, parasitic fungus-associated saprotrophs, coprophilous fungus–plant saprotrophs–wood saprotrophs, and other nutrient types. These results intuitively demonstrate that plant cultivation leads to an increase in plant pathogenic fungi and indicate that artificial cultivation alters the community composition of plant endophytic fungi.

### 4.3. Influence of Artificial Cultivation on the Function of Endophytes in the Tepals of L. delavayi

Artificial cultivation not only significantly affects the diversity and community composition of endophytic fungi but also significantly alters their functions. First, when comparing wild and cultivated populations, it was found that pathways with higher richness mostly involved purine or adenosine metabolic pathways, which are related to plant growth and development. This indicates that artificial cultivation significantly affects the function of endophytic fungi. Additionally, to further understand the impact of different cultivation statuses on the endophytic fungi of *L. delavayi* tepals, we compared the functions of CX and CY. Although both strains were cultured, CY was significantly more abundant in fatty acid metabolic pathways and methyl ketone synthesis pathways than was CX. The content of unsaturated fatty acids is closely related to plant resistance to plant pathogens [70], while the content of plant methyl ketones [71] is closely related to plant resistance to insect attack. However, in the CX with a greater abundance of the L-leucine degradation I pathway, we found that leucine is associated with pollen viability and germination [72,73]. As a precursor of scent, CX is also less able to attract pollinators. As a result, its reproductive capacity is reduced. In conclusion, our results indicate that CX plants grown in nutrient-rich and nutrient-suitable environments exhibit a trend toward decreased resistance, a decreased ability to resist pests and diseases, and decreased reproductive capacity. This is evidence that different artificial cultivation conditions have a significant impact on the function of endophytic fungi in *L. delavayi*.

Our results demonstrate that artificial cultivation significantly alters the composition of endophytic fungi, introducing potential pathogens such as typical genera like Cladosporium and Alternaria, which may lead to diseases such as leaf spot in plants. However, the protection provided by endophytic fungal diversity extends beyond singular mechanisms, interacting in multiple ways to establish complex symbiotic networks that enhance plant adaptability and survival. Therefore, safeguarding and promoting endophytic fungal diversity are crucial for maintaining plant health and ecosystem stability. Concurrently, while ensuring biodiversity, introducing additional endophytic fungi as needed can achieve effective biological control.

## 5. Conclusions

In summary, the analysis of endophytic fungi in *L. delavayi* tepals indicated that cultivation leads to a decrease in the salt stress resistance, pest and disease resistance, and reproductive capacity of plants. At the same time, cultivation leads to an increase in the diversity of endophytic fungi in tepals, adding new pathogenic genera such as *Monilinia* and *Phomopsis*. The significant differences in endophytic fungal communities between tepals and leaves indicate distinct selection effects of plants on different organs. Our research provides not only a more comprehensive understanding of endophytic fungi in *L. delavayi* tepals but also a reference for the protection and breeding of *L. delavayiceae* plants in the future.

## Figures and Tables

**Figure 1 microorganisms-12-01491-f001:**
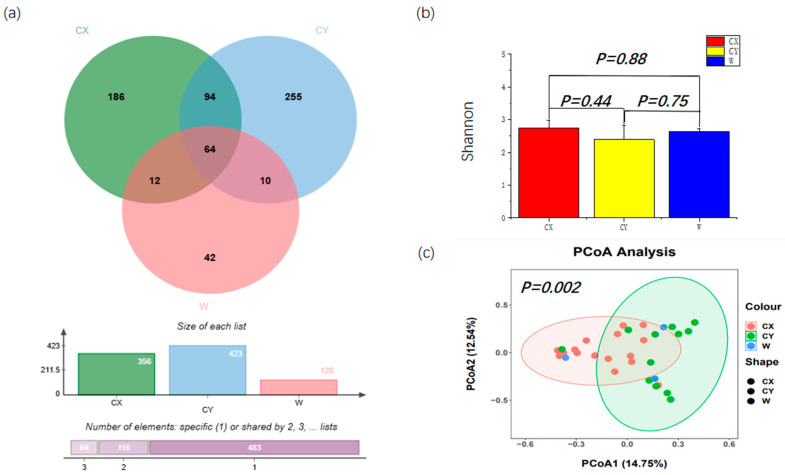
Alpha diversity and beta diversity of the tepals of *L. delavayi*. (**a**) Endophytic fungi Venn diagrams of W, CX, and CY. (**b**) Shannon indices of tepal fungal communities in W, CX, and CY. (**c**) PCoA analysis of endophytic fungal communities in W, CX, and CY. The Bray–Curtis distance matrix was used, where the horizontal and vertical percentiles represent the degree of interpretation of the samples on the first and second principal axes, respectively.

**Figure 2 microorganisms-12-01491-f002:**
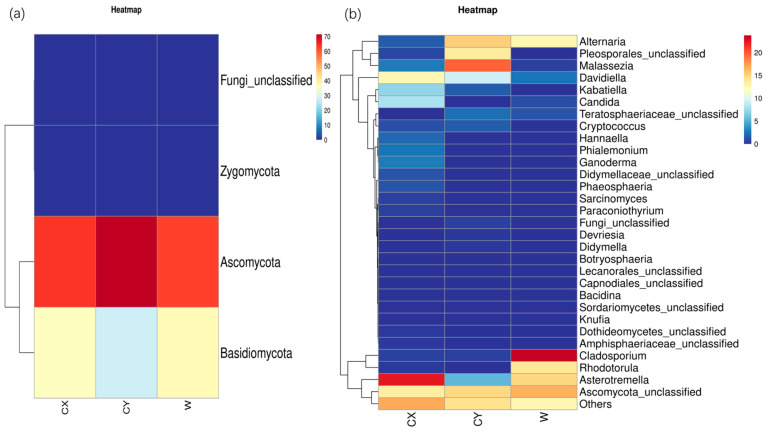
Phylum (**a**) and genus (**b**) heatmaps of tepal endophytes in W, CX, and CY. The change in endophytic fungal abundance from small to large is indicated by the gradient from blue to red. The heatmap was used to normalize the expression of the same fungus after Z value transformation.

**Figure 3 microorganisms-12-01491-f003:**
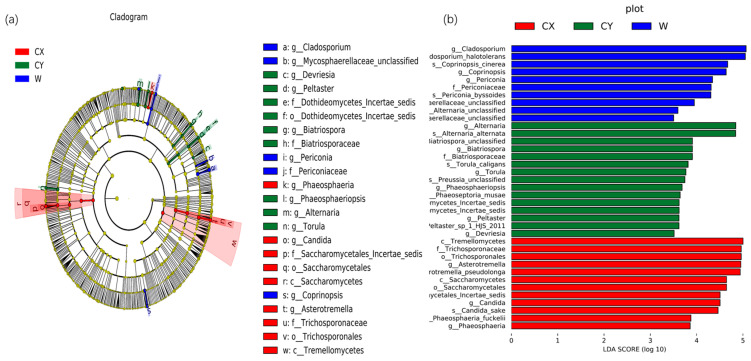
Biomarkers of endophytic fungi in the tepals of W, CX, and CY. (**a**) The LEfSe clade diagram shows the differences between W, CX, and CY at the boundary, phylum, class, order, family, genus, and species levels. Each continuous circle represents a phylogenetic level, the size of the circle represents the abundance, the different colors represent the taxa enriched by endophytic fungi in the tepals of *L. delavayi* in different cultivated and domesticated states, and the different taxa are listed on the right side of the branch diagram. (**b**) LDA bar chart showing fungal taxa, showing only taxa with LDA > 3.5.

**Figure 4 microorganisms-12-01491-f004:**
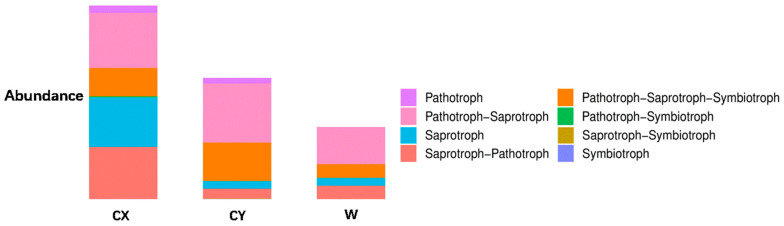
Analysis of the ecological functions of endophytic fungi in the tepals of W, CX, and CY. It is based on the three major nutritional modes of fungi.

**Figure 5 microorganisms-12-01491-f005:**
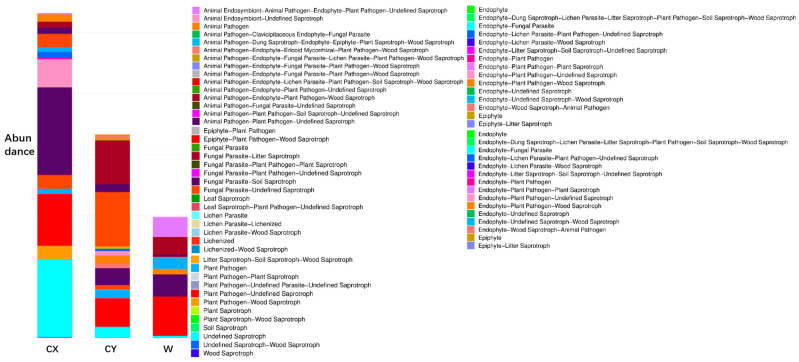
Analysis of the ecological functions of endophytic fungi in the tepals of W, CX, and CY. Classification based on 12 guild types of fungi.

**Figure 6 microorganisms-12-01491-f006:**
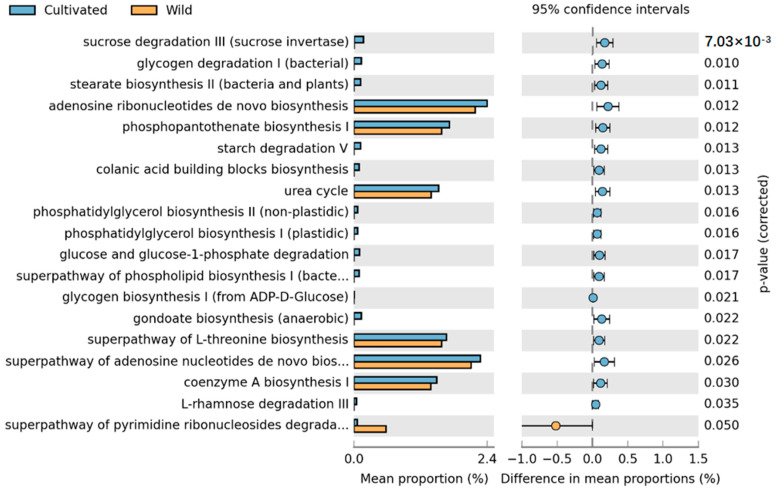
STAMP differential analysis of the prediction of the endophytic function of wild and cultivated *L. delavayi* tepals on the basis of the KEGG database (level 3). The results of the STAMP analysis in the figure are available only for the first 30 functions with *p* < 0.05 in the results of the *t* test for the difference in the two comparisons, and the results show statistically significant functions (95% confidence intervals).

**Figure 7 microorganisms-12-01491-f007:**
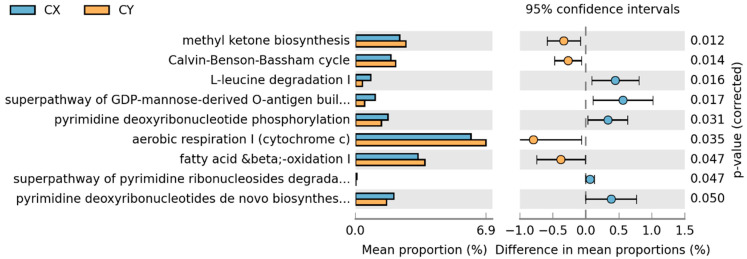
STAMP differential analysis of endophytic function prediction of tepal endophytic fungi of *L. delavayi* sinensis macrophyllum based on the KEGG database (Level 3).

## Data Availability

Online repositories containing the datasets presented in this study. The repository names and accession number(s) can be found below: https://www.ncbi.nlm.nih.gov/sra/PRJNA1109368 (URL (accessed on 20 June 2024)).

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
