# Peer review of "Characteristics of the Perianthic Endophytic Fungal Communities of the Rare Horticultural Plant Lirianthe delavayi and Their Changes under Artificial Cultivation"

_microorganisms, 2024, doi:10.3390/microorganisms12071491_

Round 1
Reviewer 1 Report
Comments and Suggestions for Authors
The authors present a descriptive account of the variability of the fungal microbiome found in the tepals of Lirianthe delavayi, depending on the degree of human alteration of the growing conditions, using bioinformatics techniques. I acknowledge all the work and effort done by the authors to study the diversity around this tree. However, while these methods are extremely useful for interpreting the results of other lab or field experiments, in my opinion, they are not sufficient on their own for publication, especially when the results are very limited and do not contribute substantively to the existing body of knowledge. The study does not offer any novel insight or methodological approach that could justify its publication. In my opinion, several aspects limit the value of the study. Firstly, the authors relied excessively on Funguild results, which would be more informative in a broader analysis. However, the authors cannot ascertain the pathogenicity or real function of the fungal strains within the analyzed plants. Additionally, a significant portion of their discussion is based on a comparison with another of their studies regarding the fungal biodiversity of Lirianthe delavayi leaves, which is neither explained nor presented before in the manuscript. Lastly, while the STAMP analysis is very useful, the predicted metabolic pathways were not confirmed to be active, nor their activity level, by any means, like qPCR. These aspects limit importantly the conclusions that can be made from the results.
I leave my comments below, hoping the authors find them useful to improve their manuscript.
Line 3: As a scientific name, Lirianthe delavayi should be italicized throughout the text.
Line 114: The authors should explain why they took a different number of samples for each group to do the analysis, as this could not be considered the ideal conditions for this kind of analysis.
Line 140: Figure 1 quality is low. Additionally, if the authors collected samples of Lirianthe delavayi, they should explain why they use pictures of Magnolia sprengeri here.
Line 259: Although Funguild can be an interesting tool for exploratory analysis, the results should be considered more cautiously than is done in the text.
Line 288: Figure 6 legend cannot be read correctly.
Line 334: Although referred to in the abstract, the analysis of the differences between endophytic fungi on leaves and tepals is not detailed in the materials and methods section nor studied in the results. Making this a fundamental part of the discussion is misleading.
Line 372: The authors should not rely on results that are not explained in the manuscript.
Line 454: The authors could discuss more exhaustively in the discussion how this study would help protect this species. Do they consider their sample size big enough to make this statement?
Comments on the Quality of English Language
Author Response
The authors present a descriptive account of the variability of the fungal microbiome found in the tepals of Lirianthe delavayi, depending on the degree of human alteration of the growing conditions, using bioinformatics techniques. I acknowledge all the work and effort done by the authors to study the diversity around this tree. However, while these methods are extremely useful for interpreting the results of other lab or field experiments, in my opinion, they are not sufficient on their own for publication, especially when the results are very limited and do not contribute substantively to the existing body of knowledge. The study does not offer any novel insight or methodological approach that could justify its publication. In my opinion, several aspects limit the value of the study. Firstly, the authors relied excessively on Funguild results, which would be more informative in a broader analysis. However, the authors cannot ascertain the pathogenicity or real function of the fungal strains within the analyzed plants. Additionally, a significant portion of their discussion is based on a comparison with another of their studies regarding the fungal biodiversity of Lirianthe delavayi leaves, which is neither explained nor presented before in the manuscript. Lastly, while the STAMP analysis is very useful, the predicted metabolic pathways were not confirmed to be active, nor their activity level, by any means, like qPCR. These aspects limit importantly the conclusions that can be made from the results.
Response: Thank you so much. The article ‘Characteristics of the perianthic endophytic fungal communities of the rare horticultural plant Lirianthe delavayi and their changes under artificial cultivation’ by yuan et al. The aim was to reveal the diversity of endophytic fungi in the tepal of magnolia and the effects of artificial cultivation and the environment on it. In this study, we did not overly rely on funguild's functional prediction results. Instead, we focused our analysis on dominant and unique fungal groups based on their abundance in measured data, which aligned with predicted functionalities. We referenced studies on yeasts of the genus Rhodotorula and fungi of the genus Colletotrichum, supplementing our findings, although we did not explicitly analyze the functions of the analyzed strains. Additionally, we conducted differential analyses of floral organs and leaves to highlight the impact of artificial cultivation on the diversity of endophytic fungi in different plant organs. Metabolic pathways were introduced as a means to understand fungal functions but were not the primary focus of the article.This gives additional insights on how the mechanisms of plant-microbe interactions are affected by human interventions, and provided a few recommendations for the conservation of the species.
Comments1:As a scientific name, Lirianthe delavayi should be italicized throughout the text.
Response1:Thank you so much. Based on your feedback, we have corrected all Lirianthe delavayi's to italics.
Comments 2:The authors should explain why they took a different number of samples for each group to do the analysis, as this could not be considered the ideal conditions for this kind of analysis.
Response 2:Thank you very much for your suggestions.We are sorry that our inaccurate statement made you misunderstand our experimental design.In general, equal sample sizes ensure unbiased comparisons between groups, thus reducing the likelihood of skewed results due to unequal sample sizes.For all tepal samples, each tepal sample consisted of three healthy flowers from an individual tree and served as one biological replicate. Due to the rarity of wild Lirianthe delavayi, and the extremely brief presence of its flowers, only three sets of biological replicates were collected. Cultivated Lirianthe delavayi found on the Yunnan University campus numbered only eight trees, while those found in the wild environment numbered six trees, all within a distance of less than 2 kilometers. Additionally, samples collected from each tree were divided into two groups based on different flowering periods. Thus, a total of 16 samples were collected from the horticultural cultivation environment and 12 samples were collected from the wild environment.
Comments 3:Figure 1 quality is low. Additionally, if the authors collected samples of Lirianthe delavayi, they should explain why they use pictures of Magnolia sprengeri here.
Response 3:Thanks a lot. We have enhanced the clarity of Figure 1.But I don't agree with your second point。We've been very familiar with Lirianthe delavayi for a long time, and we've been very careful about identifying this species.Moreover, according to the Flora of China, Magnolia sprengeri was found only in Shaanxi, Hubei, Sichuan, and other regions of China, and not in Yunnan. Our survey of the sampling site around Kunming did not find any occurrence of Magnolia sprengeri. Additionally, our analysis of the chloroplast genome of the sampled specimens confirmed them to be Lirianthe delavayi. This article has also been cited as a reference in our study. We have ample evidence to support that the specimens we sampled are indeed Lirianthe delavayi.
Comments 4:Although Funguild can be an interesting tool for exploratory analysis, the results should be considered more cautiously than is done in the text.
Response 4:Thanks to your suggestion, but I don't agree with your point,too. Funguild, as a tool that integrates fungal ecological functions, I believe, holds significant advantages for revealing the diversity of endophytic fungi in plants and illustrating the characteristics of plant endophytic fungal communities. The results it presents are closely aligned with the intended message of our article. Additionally, we have observed an increasing number of studies using this tool to analyze the diversity of endophytic fungi in plants.
Comments 5:Figure 6 legend cannot be read correctly.
Response 5:Thanks to your advice, we've recomposed the image and optimized the image to make it clearer.
Comments 6:Although referred to in the abstract, the analysis of the differences between endophytic fungi on leaves and tepals is not detailed in the materials and methods section nor studied in the results. Making this a fundamental part of the discussion is misleading.
Response 6:Thanks. Based on your feedback, we have read article clearly and made the following explanation: the main focus of this article is to reveal the impact of artificial cultivation and different habitats on the endophytic fungi in the tepals of Lirianthe delavayi. However, our laboratory has also made significant discoveries regarding the diversity of endophytic fungi in Magnolia leaves. Although our discussion is based directly on the results obtained from these two articles, which may appear somewhat crude, it does not affect the outcomes we aim to achieve. Specifically, the significant differences observed in the endophytic fungi between leaves and flower tepals under artificial cultivation are based on reliable data from these two studies. Therefore, we believe that this approach is not misleading but rather necessary.
Comments 7:The authors should not rely on results that are not explained in the manuscript.
Response 7: Thank you very much. But I'm not sure what you mean by the results in the manuscript that have not been explained.
Comments 8:The authors could discuss more exhaustively in the discussion how this study would help protect this species. Do they consider their sample size big enough to make this statement?
Response 8: Thank you so much. The paper analyzed the functions of endophytic fungi in the tepals of Lirianthe delavayi and identified dominant genera that may contribute to certain plant diseases. However, we did not discuss strategies for protecting this species in light of these findings. Following your advice, we have added relevant content to the end of the discussion section in the revised manuscript, as described below (highlighted in our revised manuscript):
"Our results demonstrate that artificial cultivation significantly alters the composition of endophytic fungi, introducing potential pathogens such as typical genera like Cladosporium and Alternaria, which may lead to diseases such as leaf spot in plants. However, the protection provided by endophytic fungal diversity extends beyond singular mechanisms, interacting in multiple ways to establish complex symbiotic networks that enhance plant adaptability and survival. Therefore, safeguarding and promoting endophytic fungal diversity are crucial for maintaining plant health and ecosystem stability. Concurrently, while ensuring biodiversity, introducing additional endophytic fungi as needed can achieve effective biological control."

Reviewer 2 Report
Comments and Suggestions for Authors
The manuscript entitled “Characteristics of the perianthic endophytic fungal communities of the rare horticultural plant Lirianthe delavayi and their changes under artificial cultivation” It focuses on analyzing and determining the endophytic fungal community of tepals in Lirianthe delavayi and how this composition affects both wild and artificial cultures. Their results showed that there is a variety of endophytic fungi in both cultivations. Furthermore, artificial cultivation exhibited altered metabolic pathways of endophytic fungi, decreasing their ability to resist pest and diseases. The experimental study is very complete.

Author Response
The manuscript entitled “Characteristics of the perianthic endophytic fungal communities of the rare horticultural plant Lirianthe delavayi and their changes under artificial cultivation” It focuses on analyzing and determining the endophytic fungal community of tepals in Lirianthe delavayi and how this composition affects both wild and artificial cultures. Their results showed that there is a variety of endophytic fungi in both cultivations. Furthermore, artificial cultivation exhibited altered metabolic pathways of endophytic fungi, decreasing their ability to resist pest and diseases. The experimental study is very complete.
The manuscript can be accepted for publication, with minor observations.
Response:Thank you very much.We have agreed to include all the changes to the text to improve the manuscript and make it more clear and readable.
Comments1:At line 160, Specify the meaning of the acronym ASV, it is also mentioned in the abstract but before this line number its meaning is not specified. This is specified on line 328, it must be specified the first time it appears in the text.
Response1:Thank you so much. Based on your feedback, we have revised the abstract to clarify the meaning of 'ASV' as well as to specify its meaning the first time it appears in the text.
Comments 2:At line 199, correct the word “w had” to “we had”
Response 2:Thank you very much for your suggestions. We have changed ‘w had’ into ‘we had’
Comments 3:Adjust the paragraph from lines 300 to 307.
Response 3:Thanks. As your suggestion, we have adjust the paragraph from lines 300 to 307.
Comments 4:At line 337, delete the reference in brackets.
rResponse 4:Thanks to your advice, we have deleted the reference in brackets.
Comments 5:At line 421, correct the word “population” to “population”
Response 5:Thank you very much. But I'm not sure what you mean.
Comments 6:At line 473, in References section, write the names of the Journals according to the journal’s instructions. The numbering is repeated in all references, please correct this.
Response 6:Thanks a lot. As your suggestion, we have partially corrected the duplicate numbering of references.
Comments 7:At line 597, add reference number 57.
Response 7:Thanks.Based on your feedback, we have read article clearly and added reference number 57.

Reviewer 3 Report
Comments and Suggestions for Authors
Lang Yuan and colleagues present an article on the composition of the endophytic fungal community in the tepals Lirianthe delavayi, complementing their previous study on the identification of endophytic fungi within leaves. Differences between wt and cultivated plants are discussed, and the influence on endophytes in the different conditions is correlated. In general, the manuscript is comprehensibly written, the experiments appear to well conducted and the data offered may prove important for people working in this field.
In order to highlight novelty, the article would really benefit from an additional figure illustrating the comparison and identified differences in the mycobiomes of leaves and petals to better complement Chapter 4.1.
Some more time must be invested in the presentation of existing Figures. For example, higher resolution of photographs in Figure 1 should be provided and some kind of map of the sample area could also be provided. In Figure 5 and especially in Figure 6, the bar graphs could become much smaller and the indexes of what corresponds to what must be presented using a bigger font size and perhaps also be placed at the right side of the figure.
Line 16-17: Please rephrase and avoid the word ‘abundant’ in this occasion.
Line 29: What do the authors mean that ‘L. delavayi is endophytic’? Please rephrase.
Author Response
Lang Yuan and colleagues present an article on the composition of the endophytic fungal community in the tepals Lirianthe delavayi, complementing their previous study on the identification of endophytic fungi within leaves. Differences between wt and cultivated plants are discussed, and the influence on endophytes in the different conditions is correlated. In general, the manuscript is comprehensibly written, the experiments appear to well conducted and the data offered may prove important for people working in this field.
In order to highlight novelty, the article would really benefit from an additional figure illustrating the comparison and identified differences in the mycobiomes of leaves and petals to better complement Chapter 4.1.
Some more time must be invested in the presentation of existing Figures. For example, higher resolution of photographs in Figure 1 should be provided and some kind of map of the sample area could also be provided. In Figure 5 and especially in Figure 6, the bar graphs could become much smaller and the indexes of what corresponds to what must be presented using a bigger font size and perhaps also be placed at the right side of the figure.
Response:We thank you very much for your suggestion "In order to highlight novelty, the article would really benefit from an additional figure illustrating the comparison and identified differences in the mycobiomes of leaves and petals".We think the text is sufficiently explains the issue, clearly articulating our main points that floral endophytic fungi diversity is lower than in leaves, and notable species differences like Rhodotorula and typical endophytic pathogenic fungi Alternaria in floral parts and so on.Thank you very much for the suggestions of figure1, figure 5 and figure 6.Based on your feedback on the figures, we have improved their clarity and made revisions accordingly in the manuscript.
Comments1:Please rephrase and avoid the word ‘abundant’ in this occasion.
Response1:Thank you so much. As your suggestion, we have changed ‘abundant’ into ‘various’
Comments 2:What do the authors mean that ‘L. delavayi is endophytic’? Please rephrase.
Response 2:Thank you very much. We have read carefully and revised it.

Round 2
Reviewer 1 Report
Comments and Suggestions for Authors
The authors present a descriptive account of the variability of the fungal microbiome found in the tepals of Lirianthe delavayi, depending on the degree of human alteration of the growing conditions, using bioinformatics techniques. I still have some caveats and questions.
- Keywords should be words that are not included in the title of the article.
- Generic names should be italicized
Regarding Response 3: Thanks a lot. We have enhanced the clarity of Figure 1.But I don't agree with your second point。We've been very familiar with Lirianthe delavayi for a long time, and we've been very careful about identifying this species.Moreover, according to the Flora of China, Magnolia sprengeri was found only in Shaanxi, Hubei, Sichuan, and other regions of China, and not in Yunnan. Our survey of the sampling site around Kunming did not find any occurrence of Magnolia sprengeri.
I was referring to the caption of Figure 1, which reads: “Habitat of wild Magnolia sprengeri in Yunnan Province (a), cultivated Magnolia sprengeri grown on campus (b), and cultivated Magnolia sprengeri grown in the wild (c).” I just wanted to know what the need was to refer to another Magnolia species here.
Regarding “Response 6:Thanks. Based on your feedback, we have read article clearly and made the following explanation: the main focus of this article is to reveal the impact of artificial cultivation and different habitats on the endophytic fungi in the tepals of Lirianthe delavayi. However, our laboratory has also made significant discoveries regarding the diversity of endophytic fungi in Magnolia leaves. Although our discussion is based directly on the results obtained from these two articles, which may appear somewhat crude, it does not affect the outcomes we aim to achieve. Specifically, the significant differences observed in the endophytic fungi between leaves and flower tepals under artificial cultivation are based on reliable data from these two studies. Therefore, we believe that this approach is not misleading but rather necessary.”
I agree with the authors that this comparison may be of interest. However, I believe that even though the "leaves data set" has already been published, the statistical analysis that allows the authors to state on line 338 that "there were significant variations in endophytic fungi between leaves and tepals" has not been published and is not included in this manuscript. These results need to be published so that their validity can be independently assessed. I consider that stating that "Specifically, the significant differences observed in the endophytic fungi between leaves and flower tepals under artificial cultivation are based on reliable data from these two studies" without explaining how that comparative analysis was made is misleading. Could the authors explain in the Materials and Methods section the procedure for this comparison?
Additionally, even with these data, I also think that the results of comparing the structure of fungal communities of different organs from samples not necessarily taken from the same trees and collected a year and a half apart should be taken with more caution. With this time difference, even the structure of the fungal communities in the leaves could have changed completely.
Author Response
Response to Referee 1 (round 2)
1.The authors present a descriptive account of the variability of the fungal microbiome found in the tepals of Lirianthe delavayi, depending on the degree of human alteration of the growing conditions, using bioinformatics techniques. I still have some caveats and questions.
- Keywords should be words that are not included in the title of the article.
- Generic names should be italicized
Reponse 1:Thank you very much ,Based on your suggestions, we have made modifications to the keywords and italicized the generic names.
2.I was referring to the caption of Figure 1, which reads: “Habitat of wild Magnolia sprengeri in Yunnan Province (a), cultivated Magnolia sprengeri grown on campus (b), and cultivated Magnolia sprengeri grown in the wild (c).” I just wanted to know what the need was to refer to another Magnolia species here.
Reponse 2:Regarding your the caption of Figure 1, we apologize for my oversight in the figure caption error, and we have corrected it,and according another referee’s suggestion, we have moved Figure 1 to the supplementary material.
3.I agree with the authors that this comparison may be of interest. However, I believe that even though the "leaves data set" has already been published, the statistical analysis that allows the authors to state on line 338 that "there were significant variations in endophytic fungi between leaves and tepals" has not been published and is not included in this manuscript. These results need to be published so that their validity can be independently assessed. I consider that stating that "Specifically, the significant differences observed in the endophytic fungi between leaves and flower tepals under artificial cultivation are based on reliable data from these two studies" without explaining how that comparative analysis was made is misleading. Could the authors explain in the Materials and Methods section the procedure for this comparison?
Additionally, even with these data, I also think that the results of comparing the structure of fungal communities of different organs from samples not necessarily taken from the same trees and collected a year and a half apart should be taken with more caution. With this time difference, even the structure of the fungal communities in the leaves could have changed completely.
Response 3:Thank you very much, According your suggestion, we have removed section 4.1. However, as the leaf and the perianth are aerial organs, and their surroundings are similar, except that the perianth has a shorter survival time than the leaf. Based on our previous studies on leaves, we have still included two important discussion in the revised article, categorizing them under sections 4.1 and 4.2, as described below (highlighted in our revised manuscript):
‘Coincidentally, previous studies in the laboratory on endophytic fungi in leaves [31] have shown that artificial cultivation can lead to reduced diversity of endophytic fungi in plant leaves. However, our results show the opposite situation, suggesting that this may be due to antagonistic substances, such as floral scents [45] or floral secretions [46,47], which reduce the abundance of endophytic fungi in the petals compared to that in the leaves.’
‘However, the results exhibited by the Alternaria differ significantly from previous laboratory studies on endophytic fungi in leaves.Through our understanding, numerous contradictory pieces of evidence regarding the impact of artificial plant cultivation on microbial diversity suggest that potential artificial cultivation syndrome mitigation depends on plant species [61,62]. Moreover, studies have shown that plants likely control the assembly of fungal microbial communities as a protective mechanism against fungal pathogens [63,64]. Based on these findings, we believe that the significant difference in endophytic fungi between the nutrient organs and reproductive organs of L. delavayis may be the result of plant selection effects and limited fungal dispersal [28].’

Reviewer 3 Report
Comments and Suggestions for Authors
The revised manuscript has been improved.
Since the authors have decided not to modify Figure 1, I would suggest that this is moved to the supplement. In addition, please correct the figure legend of Figure 1 (I guess cultivated Magnolia sprengeri grown on campus is shown in panel c and not b).
Author Response
Response to Referee 3 (round 2)
The revised manuscript has been improved.
Since the authors have decided not to modify Figure 1, I would suggest that this is moved to the supplement. In addition, please correct the figure legend of Figure 1 (I guess cultivated Magnolia sprengeri grown on campus is shown in panel c and not b).
Response:Thank you very much, we have moved Figure 1 to the supplementary material based on your suggestion, and we have corrected the errors in the legend.
